# Reassessing Whether Biodegradable Microplastics Are Environmentally Friendly: Differences in Earthworm Physiological Responses and Soil Carbon Function Impacts

**DOI:** 10.3390/antiox14101197

**Published:** 2025-10-01

**Authors:** Yuze Li, Dongxing Zhou, Hongyan Wang, Wenfei Zhu, Rui Wang, Yucui Ning

**Affiliations:** College of Resources and Environment, Northeast Agricultural University, Harbin 150030, China; s230201030@neau.edu.cn (Y.L.); zhouboshi@neau.edu.cn (D.Z.); s240201055@neau.edu.cn (H.W.); s240201059@neau.edu.cn (W.Z.); s240202099@neau.edu.cn (R.W.)

**Keywords:** microplastics, *Eisenia fetida*, haplic phaeozem, factor analysis, principal component analysis, path analysis

## Abstract

Biodegradable plastics are not a primary solution to plastic pollution, and empirical evidence on whether they are environmentally friendly remains lacking. In this study, we systematically compared the toxic effects of traditional microplastics (polypropylene, PP; polystyrene, PS) with biodegradable microplastics (polylactic acid, PLA; polyhydroxyalkanoates, PHA) on the haplic phaeozem ecosystem. Through mathematical modeling analysis, it was found that earthworms initially rely on antioxidant enzymes to resist stress, mid-term activation of detoxifying enzymes to repair damage, and maintaining physiological balance through metabolic regulation and immune enhancement in later stages. We elucidated their mechanism differences: PLA and PP caused severe damage to the antioxidant system and cell membrane, with PLA mainly relying on POD to clear peroxides and PP relying on GST. In addition, PLA and PS can induce early neurotoxicity (AChE), while PHA induces late neurotoxicity. Furthermore, this study provides direct evidence proving that biodegradable microplastics are not environmentally friendly by breaking through the one-way research framework of “microplastic biotoxicity” and innovatively constructing a path analysis model that links biological physiological responses with soil ecological functions. We also provide a scientific basis to evaluate the ecological risks of microplastic pollution in soil and the whether biodegradable plastics are truly environmentally friendly.

## 1. Introduction

Since the first successful synthesis in 1907, plastics have been widely utilized across various industries due to their low production cost, high plasticity, and stable chemical properties [1]. However, a large amount of plastic waste is improperly managed. Under the influence of ultraviolet radiation, microbial degradation, mechanical abrasion, and other environmental factors, unrecycled plastics gradually fragment into particles smaller than 5 mm in diameter [2], known as microplastics (MPs). Among these, polyethylene (PE), polypropylene (PP), and polystyrene (PS) are the most common [3]. With the long-term and continuous accumulation of these microplastics in the soil, they will not only have a negative impact on soil structure, but also hinder nutrient cycling, reduce water retention capacity, and pose serious threats to soil biota [4], making them a pressing global environmental concern [5].

To mitigate the environmental risks associated with traditional microplastics, the draft of the Global Plastics Treaty [6] indicates that biodegradable plastics may serve as a potential means to end plastic pollution. This perspective, coupled with the continuous growth of human demand for plastics, has led to the global production of biodegradable plastics reaching 2.47 million tons in 2024, with a projected increase to 5.73 million tons by 2029 [7]. Biodegradable microplastics ultimately degrade carbon dioxide and water, and are therefore labeled as “eco-friendly” materials [8]. However, whether biodegradable microplastics are truly environmentally friendly, especially in soil ecosystems, lacks empirical evidence. Considering that pollution remediation is very difficult, the EU is very cautious about the use of biodegradable materials. Therefore, the Single-Use Plastics Directive [9] passed by the European Union in June 2019 imposes restrictions on some oxygen-containing biodegradable plastic products. The implementation guidance document for this directive, released in May 2021, further clarifies that biodegradable plastics or biobased plastics still belong to the category “plastics” and cannot be marked as “biodegradable” when used in disposable products. This has also raised concerns about the concept of “biodegradability”.

Earthworms are among the most common invertebrates in soil ecosystems. Through their feeding and burrowing activities, they facilitate the decomposition of organic matter, enhance soil porosity, and improve the soil’s capacity to retain water and nutrients. As such, they are often referred to as “ecosystem engineers” of the soil [10,11]. Due to their widespread distribution, low trophic level, and high exposure potential to environmental contaminants, earthworms are commonly used as bioindicators for assessing soil pollution and environmental quality, effectively reflecting the toxicological impacts of pollutants on terrestrial ecosystems [12].

However, earthworms have numerous response indicators after stress, and when using traditional biostatistics for data analysis, it is necessary to encompass the entire physiological and biochemical processes of the organism. As a result, the experimental content is numerous, the process is complex, and the credibility of the data analysis results is often poor [13]. At the same time, a single mathematical model suffers from excessive loss of information during data extraction, loss of biological significance of data in analysis results, and inaccurate judgment matrices due to the excessive increasing workload [14]—and is sometimes even unable to correctly explain the biological significance of the model [15]. Therefore, in the ecological toxicology research of earthworms, there is an urgent need for a combined model of principal component analysis and factor analysis to achieve accurate and scientific screening of biomarkers, in order to assist in the ecological safety evaluation and bioremediation research of polluted soil.

In this context, the present study employed *Eisenia fetida* as the model organism to investigate the oxidative stress responses induced by exposure to four types of microplastics: PP, PS, PLA (polylactic acid), and PHA (polyhydroxyalkanoates). An innovative “progressive factor analysis principal component analysis model” is used to identify key sensitive oxidative stress biomarkers of earthworms exposed to microplastics, thereby elucidating their survival strategies in soil contaminated with microplastics. In addition, the path analysis method is used to link biological physiological responses with soil ecological functions to clarify whether biodegradable microplastics are truly environmentally friendly.

## 2. Materials and Methods

### 2.1. Characterization of Microplastics

The microplastics used in this study, PP, PS, PLA, and PHA, were 200 mesh (75 μm ± 10 μm) powders supplied by Zhongxin Plastic Co., Ltd., Guangzhou, China. The chemical structure of the microplastics was confirmed using a Raman spectrometer (inVia, Renishaw, Gloucestershire, UK) with a laser wavelength of 785 nm and a wavenumber range of 200–3000 cm^−1^. After the microplastics were sputter-coated with gold, their morphologies were observed using a scanning electron microscope (SU8010, Hitachi, Tokyo, Japan).

### 2.2. Test Reagents

Phosphate-buffered saline (PBS, dry powder) was purchased from Coolable Technology Co., Ltd., Beijing, China (analytical grade). Each packet was diluted with ultrapure water to a final volume of 1 L, with a pH of 7.2–7.4. The final buffer composition included 137 mM NaCl, 2.7 mM KCl, 10 mM Na_2_HPO_4_, and 2 mM KH_2_PO_4_. Medical-grade normal saline was obtained from local pharmacies.

### 2.3. Test Soil

The test soil was black soil that was collected from the experimental fields of Northeast Agricultural University (pH: 7.65; organic matter: 17.36%); total nitrogen: 0.901%). Fresh topsoil samples (0–20 cm depth) were collected, and plant debris, stones, and other impurities were removed. The soil was air dried and sieved through a 5 mm mesh. Semi-decomposed cow dung was used as an earthworm feed source, mixed with the soil at a ratio of 7:3 (*w*/*w*) (black soil:cow dung) (microplastic: 0.05 mg/kg, cadmium; lead; nickel: <0.01 mg/kg). The moisture content was adjusted to 40% of the soil’s maximum water holding capacity using deionized water, following OECD guidelines [16,17]. The cow dung was prepared by natural air drying, grinding, and sieving through a 5 mm mesh.

### 2.4. Tested Earthworms

The test organism, *Eisenia fetida*, was sourced from a national ecological farm. Healthy adult earthworms, over 2 months old, with well-developed clitella and weighing between 300 and 500 mg, were selected. Prior to the experiment, earthworms were rinsed with clean water to remove surface debris and residual feed. After drying with filter paper, they were placed on moist filter paper in a dark environment for 24 h to purge intestinal contents.

### 2.5. Experimental Design

Based on the results of preliminary experiments and previous research findings [18,19,20,21], four microplastic types (PP, PS, PLA, and PHA) were tested at concentrations of 0, 100, 500, 1000, and 1500 mg/kg. Each treatment was replicated five times. The exposure test was conducted in a 1000 mL wide-mouthed bottle, each containing 120 g of treated test soil mixed thoroughly with the respective concentration of microplastics. Five depurated earthworms were placed in each flask. The flasks were covered with plastic film to prevent moisture loss and incubated in a controlled artificial climate chamber at (20 ± 2) °C, 80% relative humidity, and in complete darkness.

### 2.6. Sample Collection

According to the sampling requirements of weekly intervals in the OECD guidelines [16], the experimental period lasted 35 days. One earthworm was randomly selected from each flask every 7 days (take a total of 5 earthworms each time). After collection, the earthworm’s surface was rinsed with deionized water and gently dried using filter paper. The sample was then transferred to a glass homogenizer, and phosphate-buffered saline (PBS, pH 7.3) was added at a ratio of 1:9 (*w*/*v*, accurate to 0.0001 g) for homogenization. The resulting homogenate was transferred to a centrifuge tube and centrifuged at 4000 r/min for 10 min at 4 °C. The supernatant was collected and stored at −20 °C for subsequent biochemical analysis. An amount of 10 g soil was taken from each bottle every 7 days, air-dried naturally, ground and then sieved through a 200 mesh sieve for the determination of total soil carbon.

### 2.7. Determination of Enzyme Indexes

Frozen supernatant samples were thawed at +4 °C prior to analysis. The following biochemical parameters were determined using standard protocols: the total protein content was determined by Coomassie Brilliant Blue Colorimetry [22]; peroxidase (POD) activity was assessed based on the catalytic reaction of POD with hydrogen peroxide [23]; superoxide dismutase (SOD) activity was determined via its specific inhibitory effect on superoxide anion radicals [24]; glutathione peroxidase (GPX) activity was measured using 2-thio-2,4-dinitrobenzoic acid colorimetry [25]; glutathione-S-transferase (GST) activity was evaluated by 1-chloro-2,4-dinitrobenzene colorimetry [26]; catalase (CAT) activity was determined using ammonium molybdate colorimetry [27]; malondialdehyde (MDA) content was measured via the thiobarbituric acid (TBA) method [28]; acetylcholinesterase (AChE) activity was analyzed using 1,3,5-trinitrobenzene colorimetry [29]. Soil total carbon (TC) was measured using the combustion method [30].

### 2.8. Data Analysis

#### 2.8.1. Factor Analysis

Factor analysis is a multivariate statistical method that reduces dimensionality by transforming multiple index variables into a few comprehensive factors [14,31]. To minimize the impact of variability among oxidative stress indicators, the original data were normalized and expressed as the percentage change in each test group relative to the sum across all groups:$$U=\frac{x_{s}}{\sum_{s=1}^{t}x_{st}}\times 100\%$$

Here, *x_S_* represents the oxidative stress indicators, including POD, SOD, CAT, MDA, AChE, GPX, and GST; *t* represents the number of samples.

After dimensionality reduction, the *n* biological markers can be represented as a linear combination of *m* common factors (m < n), denoted as F1,F2,…Fm. The construction process of the model refers to the research of Zhou et al. [32].

All statistical analyses were conducted using SPSS software version 19.0.

#### 2.8.2. Principal Component Analysis

Principal component analysis (PCA) [33] is a multivariate statistical method used to explore correlations among multiple variables. PCA identifies the internal structure of complex datasets by transforming them into a smaller set of uncorrelated principal components, thereby simplifying data interpretation [34]. In this study, PCA was applied to analyze the oxidative stress response of earthworms under microplastic exposure. The index matrices of the experimental group and the control group were denoted as *X_f_* and *X*, respectively:$$X_{f}=\left\{\begin{array}{cccc} x_{11} & x_{12} & \cdots & x_{1g}\\ x_{21} & x_{22} & \cdots & x_{2g}\\ \cdots & \cdots & \cdots & \cdots \\ x_{n1} & x_{n2} & \cdots & x_{ng} \end{array}\right\}\;X_{0}=\left\{\begin{array}{cccc} x_{01} & x_{02} & \cdots & x_{0n}\\ x_{01} & x_{02} & \cdots & x_{0n}\\ \cdots & \cdots & \cdots & \cdots \\ x_{01} & x_{02} & \cdots & x_{0n} \end{array}\right\}$$

The variable sample matrix is then obtained as: $∆X=X_{f}-X_{0}$

In this context, *f* is the stress time; *g* is the number of samples, which is the concentration of microplastic stress; n is the number of oxidative stress indicators, including POD, SOD, CAT, MDA, AChE, GPX, and GST, measured in earthworms exposed to microplastic-contaminated soil. To reduce dimensionality while minimizing squared error in the sample set, coordinate transformation was applied using the Jacobi method to compute the orthogonal transformation matrix. A total of *m* principal components were then extracted, where *m < n*. The Z-score method is used for standardization (Appendix A for detailed process)$$C_{\eta =1}^{n}=\varepsilon_{\eta 1}X_{j1}+\varepsilon_{\eta 2}X_{j2}+\cdots +\varepsilon_{\eta n}X_{jn},\;S_{\eta =1}^{n}=\left|\sum_{\eta =1}^{m}C_{\eta j}^{n}\xi_{\eta }\right|$$

All data processing and statistical analysis were conducted using SPSS software version 19.0.

#### 2.8.3. Path Analysis

After the occurrence of microplastic stress, the oxidative stress index in earthworms is used as the independent variable $X_{i}$ and the total carbon content of soil was the dependent variable Y. A path analysis model was constructed in order to analyze the relationship between biological physiological responses and soil ecological functions in microplastic-polluted environments, where

$r_{ij}$ represents the simple correlation coefficient between $x_{i}$ and $x_{j}$; $r_{ij}$ represents the correlation coefficient between $x_{i}$ and y; $P_{ij}$ is the direct path coefficient, which represents the magnitude of $x_{i}$ directly acting on Y when other variables are fixed. Additionally, $r_{ij}$ was decomposed into the following system of equations:$$\left\{\begin{array}{c} r_{1y}=P_{1y}+P_{2y}r_{12}+P_{3y}r_{13}+\cdots +P_{ny}r_{1n}\\ r_{2y}=P_{2y}+P_{1y}r_{21}+P_{3y}r_{23}+\cdots +P_{ny}r_{2n}\\ \cdots \\ r_{iy}=P_{iy}+\sum_{j\neq 1}P_{jy}r_{ij}\left(i=1,2,\cdots ,n\right)\\ \cdots \\ r_{ny}=P_{ny}+P_{1y}r_{n1}+P_{2y}r_{n2}+\cdots +P_{\left(n-1\right)y}r_{n\left(n-1\right)} \end{array}\right.$$

The remaining diameter coefficient must be ≤0.25.

## 3. Results and Discussion

### 3.1. Characteristics of Microplastics

As shown in Figure 1, the surfaces of PP and PS are smooth and flat, with intact edges, and without obvious pores, rough protrusions or agglomeration phenomena, and their overall morphologies are neat and concise. A part of the PLA surface has a rough texture, and even has porous or uneven structures. PHA presents rod-shaped or fibrous aggregation structures—not in granular form, but forms clusters composed of slender structures. Raman spectroscopy was used to identify the characteristic peaks of microplastics in the range of 200–3000 cm^−1^. As shown in Figure 2, PP exhibited characteristic peaks at 809, 842, 974, 1153, 1331 and 1461 cm^−1^, corresponding to distinct molecular vibrations [35]. Specifically, the peaks at 809 and 842 cm^−1^ are attributed to C-H rocking vibrations, 974 cm^−1^ to asymmetric C-C stretching, 1153 and 1331 cm^−1^ to C-C stretching vibrations, and 1461 cm^−1^ to C-H bending vibrations [36]. PS showed characteristic peaks at 621, 792, 1002, 1156, 1445, and 1604 cm^−1^, respectively [37]. Among these, the peaks at 621, 792 and 1445 cm^−1^ correspond to C-C stretching vibrations, while the peak at 1604 cm^−1^ is associated with C-H stretching in the benzene ring [38,39]. Notably, the peak at 1002 cm^−1^, representing the ring breathing mode of the benzene ring, was the most prominent and is commonly used as a diagnostic peak for PS identification [40].

For polylactic acid (PLA), a peak at 411 cm^−1^ was assigned to C=O deformation vibration, and the peak at 874 cm^−1^ was due to the (C-COO) stretching of the polymer chain repeat units. Additional peaks at 1044, 1129, and 1453 cm^−1^ corresponded to C-CH_3_ stretching, CH_3_ in-plane bending, and CH_3_ asymmetric bending, respectively. The peak at 1767 cm^−1^ indicated C=O stretching, while the strong peak at 2948 cm^−1^ was attributed to symmetric CH_3_ stretching [41]. PHA showed characteristic peaks at 839, 1059, 1366, and 1725 cm^−1^. The peak at 839 cm^−1^ was related to C-COO stretching, 1059 cm^−1^ to C-C stretching, and 1366 cm^−1^ to C-H rocking vibrations. The peak at 1725 cm^−1^, associated with C=O stretching, served as a key marker for identifying PHA [42].

### 3.2. Physiological Response of Earthworms Under Microplastic Stress

#### 3.2.1. Oxidative Stress Effects in Earthworms

As shown in Figure 3, under PP stress, the activities of most antioxidant enzymes in *Eisenia fetida* peaked at a concentration of 500 mg/kg and subsequently declined with increasing concentration. Specifically, at 500 mg/kg, the maximum SOD activity was observed on the 28th day of stress, at 97.62 U/mg prot; the maximum GPX activity was detected on the 21st day of stress, at 183.66 U/mg prot; and the maximum AChE activity was noted on the 21st day of stress. At this concentration (500 mg/kg), AChE activity gradually decreased with the passage of time. Under different stress concentrations and exposure times, the CAT activity of the PP treatment group was lower than that of the control group (CK).

Under PS stress (Figure 3), enzyme activities generally increased at 1000 mg/kg, but declined significantly at 1500 mg/kg. POD and SOD reached their highest activities, 66.02 and 113.16 U/mg prot, respectively, on day 7 under 1000 mg/kg stress, followed by a gradual decrease at higher concentrations. GST activity peaked at 63.16 U/mg prot on the 14th day under 1000 mg/kg of PS stress. However, GST activity dropped sharply to 4.89 U/mg prot at 1500 mg/kg.

For PLA stress (Figure 3), most enzyme activities reached their maximum at low concentrations (100 and 500 mg/kg). For example, the peak activities of POD, SOD, and AChE were observed at 100 mg/kg on the 7th day of PLA stress. However, with increasing PLA stress concentration, the activities of POD, SOD, and AChE exhibited a decreasing trend.

Under PHA stress (Figure 3), most enzyme activities reached their peak at 1000 mg/kg of PHA stress. The activities of CAT, SOD, AChE, and GST reached their maximum values at 1000 mg/kg of PHA stress (1.2, 43.65, 12.93, and 47.44 U/mg prot, respectively), and then decreased with increasing PHA concentration. With the passage of time under this concentration (1000 mg/kg of PHA stress), the activities of CAT, AChE, and GST gradually decreased. In contrast, SOD activity exhibited a trend of first decreasing and then increasing at this concentration.

Overall, except for GST activity under PLA stress and GPX activity under PHA stress, all other enzyme activities under PS, PLA, and PHA stress were lower than those in the CK. Both traditional and biodegradable microplastics could inhibit the enzyme activities in earthworms through oxidative stress [43].

#### 3.2.2. Screening of Sensitive Oxidative Stress Indicators

A factor analysis model was constructed to evaluate the comprehensive eigenvalues, contribution rates, and factor loadings of oxidative stress indicators under different microplastic types and exposure durations. Based on this analysis, the key oxidative stress biomarkers in earthworms were identified (Table 1). Upon exposure to microplastic stress, large quantities of O_2_^−^ were generated in *Eisenia fetida*, disrupting the organism’s oxidative stress balance [44]. As shown in Table 1, the total protein content an indicator of metabolic activity fluctuated significantly under oxidative stress. SOD, which catalyzes the dismutation of O_2_^−^ into H_2_O_2_ and molecular oxygen (O_2_), plays a critical initial role in the antioxidant defense system [45]. Since H_2_O_2_ remains biologically harmful, it must be further decomposed. CAT and POD are key enzymes responsible for the breakdown of H_2_O_2_ into water and oxygen [46]. Accordingly, during the early phase of microplastic stress (day 7), SOD, POD, and CAT were identified as the most sensitive oxidative stress biomarkers (Table 1).

Reactive oxygen species (ROS) also induced lipid peroxidation, resulting in the formation of MDA, a commonly used biomarker for assessing oxidative damage to cell membranes [47]. In this study, MDA was identified as the primary oxidative stress indicator under PP exposure on days 14 and 28. As shown in Figure 3, MDA levels increased initially with stress duration, peaking on day 28 before declining. These results suggest that lipid peroxidation began on day 14 and intensified over time, reaching a maximum on day 28, followed by a potential recovery or enzyme-mediated mitigation. Similar findings have been reported in *Eisenia fetida* exposed to LDPE and PS, where oxidative lipid damage was elevated [48]. The subsequent decline in MDA content may be attributed to the activity of antioxidant and detoxification enzymes, particularly glutathione S-transferase (GST), which plays a dual role in cellular defense [49,50]. GST facilitates the conjugation of glutathione (GSH) with electrophilic substrates to form more water soluble and less toxic derivatives, which are readily excreted or metabolized. GST also scavenges reactive oxygen species and lipid peroxidation products, thereby preserving membrane integrity [51]. As shown in Table 1, GST emerged as the dominant oxidative stress biomarker on day 35, indicating that prolonged PP exposure activated GST-mediated detoxification pathways. With extended stress duration, the toxicity of PP progressively intensified, leading to a collapse of the antioxidant defense system and extensive lipid peroxidation. In response, *Eisenia fetida* activated detoxifying enzymes like GST to counteract PP-induced toxicity [32]. Similar enzyme-mediated defense mechanisms have also been observed in studies investigating the effects of heavy metals on earthworms [45].

GPX is an important antioxidant enzyme that catalyzes the redox reaction between GSH and peroxides, such as H_2_O_2_ and superoxide radicals [52]. GPX activity has also been associated with the growth and reproductive capacity of earthworms [53]. As shown in Table 1, after 14 days of PLA exposure, GPX was identified as a sensitive oxidative stress biomarker, suggesting that the growth and reproduction of *Eisenia fetida* were affected during this period. This finding is consistent with the results of Shu et al. [53], who reported that short-term microplastic exposure could elevate GPX activity, thereby inhibiting growth and reproduction. On day 14, GPX had a factor loading value of 0.856, ranking second only to GST (loading = 0.994), indicating a strong response. However, as the stress duration increased, the correlation or “companion” relationship between GPX and GST diminished. This may be due to the antagonistic interaction between the two enzymes, as they compete for the common substrate GSH [54]. GPX activity is closely regulated by GSH availability, and only with sufficient GSH supplementation can GPX function optimally [55]. As shown in Figure 3, GPX activity increased again after 28 and 35 days of PLA exposure, suggesting a delayed but persistent oxidative response. Additionally, GPX helps protect the integrity and functionality of cell membranes from oxidative damage [56]. Interestingly, during this later stage, POD was identified as a key oxidative stress indicator, and TP content showed a strong negative loading (−0.997 for POD and −0.968 for TP), indicating a high level of peroxide accumulation and a decline in antioxidant capacity. These results suggest that both PLA and PP exposure caused significant damage to the antioxidant defense system and cell membrane integrity in earthworms. However, the mechanisms differed: PLA primarily triggered a POD-mediated response to peroxides, while PP induced a GST-dominant detoxification pathway.

AChE is a critical enzyme in neural signal transmission, responsible for degrading acetylcholine (ACh) to reduce neural excitation. AChE is widely regarded as one of the most effective biomarkers to evaluate neurotoxicity caused by environmental pollutants [57]. Elevated AChE activity can lead to acetylcholine depletion, impairing cholinergic signal transmission and adversely affecting cognition and memory [58]. In this study, AChE was identified as a sensitive oxidative stress biomarker from day 14 to day 35 under PS exposure (Table 1). However, Table A2 shows that as early as day 7, AChE exhibited a strong correlation with POD (factor loading = 0.927), and Figure 3 confirms a gradual increase in AChE activity beginning on day 14. These findings suggest that PS exposure triggered early-stage neurotoxicity in earthworms, which progressively intensified over time.

A similar response was observed under PLA exposure. At day 7, AChE exhibited a strong correlation with CAT (factor loading = 0.976), indicating early neural disruption. Previous studies also reported significant increases in AChE activity in earthworms exposed to PLA, with observable neurotoxic effects [59]. Compared to PP and PHA stress, PS and PLA induced earlier and more pronounced increases in AChE activity, suggesting that these microplastics may cause greater neurotoxicity during the early stages of exposure. The summary is shown in Figure 3. According to Table A1 and Table A2, on day 21 of PHA exposure, AChE and MDA were identified as sensitive oxidative stress biomarkers, with factor loadings of 0.998 and 0.996, respectively. These results indicate that lipid peroxidation had occurred in earthworms, leading to cellular damage. The pronounced oxidative stress response reflected by elevated AChE activity may be attributed to lipid peroxidation in neural tissues, particularly the rupture of presynaptic vesicles containing neurotransmitters. This can result in an excessive accumulation of neurotransmitters in the synaptic cleft, subsequently stimulating increased AChE activity to degrade surplus acetylcholine [60]. Therefore, high AChE loadings of 0.900 and 0.871 were observed during days 28 to 35 under PHA stress, suggesting that PHA-induced neurotoxicity became more pronounced in the late stages of exposure (Figure 4).

Under PS exposure, SOD was identified as a sensitive oxidative stress indicator between days 21 and 28 (Table 1). However, Table A2 shows that TP content exhibited the highest negative loadings during this period, with values of −0.963 and −0.897, respectively. This may be closely related to the immune response of the earthworms. Previous studies by Yue et al. and Zhou et al. demonstrated that SOD plays a key role in enhancing phagocyte activity and systemic immune defense [14,61]. Thus, during this stress phase, metabolic dysfunction likely occurred in earthworms, with impaired protein synthesis and a shift in energy metabolism towards protein catabolism. In response, SOD activity increased to enhance immune function, maintain physiological homeostasis, and activate detoxification enzymes and neural regulation mechanisms. These coordinated responses help the organism recover from stress-induced damage. Similar physiological responses have been observed under cadmium exposure in earthworms [17].

### 3.3. Relationship Between the Physiological Response of Earthworms and Soil Ecological Functions Under Microplastic Stress

After standardizing the data $S_{\eta }^{n}$, an oxidative stress profile was constructed with microplastic concentration as the horizontal axis (Figure 5A). The results showed that oxidative stress responses varied across microplastic types and concentrations. Under exposure to conventional microplastics, PP exhibited a fluctuating trend of increase-decrease-increase, with the highest oxidative stress observed at 500 mg/kg. In contrast, PS followed a decrease–increase–decrease pattern, peaking at 100 mg/kg. In both cases, short-term exposure (7 and 14 days) resulted in higher oxidative stress levels than long-term exposure, suggesting that earthworms rapidly activated their antioxidant defense systems under low-concentration, short-term stress.

However, with prolonged exposure and increased concentrations, oxidative stress levels gradually declined, indicating potential suppression or exhaustion of the antioxidant defense system, which may result in oxidative damage. This reduction in defense capacity could be linked to the accumulation of microplastics in the gastrointestinal tract, leading to intestinal injury, shifts in gut microbiota, and subsequent disruption of antioxidant functions [21]. Similar findings have been reported in earthworms exposed to polyethylene microplastics (250–1000 μm) and polystyrene microplastics (5 μm and 20 μm), which induced oxidative stress and physiological disruption [19,62]. Under biodegradable microplastic exposure (PLA and PHA), oxidative stress patterns were relatively similar. Both exhibited the highest stress response at 100 mg/kg, with minimal changes in oxidative stress levels across increasing concentrations. This indicates that exposure to PLA in soil can inhibit the growth of earthworms, lead to a decrease in earthworm biomass [63], and also trigger oxidative stress in the body, causing DNA damage [64].

Based on the changes in total soil carbon under different microplastic stresses shown in Figure 5B, it can be concluded that under traditional microplastic stress, the content of total soil carbon is relatively high. This may be due to the fact that soil structure (e.g., porosity and aggregate stability) is a key factor determining soil carbon sequestration capacity [65]. The decomposition process of microplastics indirectly regulates the stability of organic matter by altering soil structure [66]. Previous studies have confirmed that traditional microplastics (e.g., PE and PVC) can increase the proportion of small-particle aggregates in soil. These small-particle aggregates mainly immobilize organic carbon through chemical reactions, which significantly reduces the decomposition rate of organic carbon and thus is more conducive to long-term soil carbon sequestration [67]. At the same time, the hydrophobicity of PP and PS particles can reduce the hydraulic conductivity of soil capillary pores, decrease the loss of organic matter via water leaching, and further promote the accumulation of soil organic matter [68].

The decomposition of biodegradable microplastics (e.g., PLA and PHA) exerts a “dynamic impact” on soil structure. In the initial stage, similar to traditional microplastics, biodegradable microplastics promote the formation of small-particle aggregates. However, as their degradation progresses, these biodegradable microplastics continue to release soluble organic matter, which further binds soil particles and gradually forms large-particle aggregates [69]. The carbon immobilization by large-particle aggregates mainly relies on physical encapsulation, and their turnover rate is relatively fast. Compared with the chemically stable carbon immobilization by small-particle aggregates, their long-term carbon sequestration capacity is weaker [70]. Another study has confirmed that adding polyester (PET) ultrafine fibers (a type of non-biodegradable microplastics) to soil significantly reduces soil bulk density and the number of water-stable aggregates, further reflecting the potential damage of such non-biodegradable microplastics to soil structure [71]. In addition, earthworm activity accelerates the degradation process of biodegradable microplastics and promotes their release of more soluble organic matter, and these soluble organic compounds are more prone to leaching and loss with soil water, ultimately leading to a decrease in total soil carbon content [72].

Similarly to traditional microplastics, the antioxidant defense system of earthworms can effectively mitigate oxidative stress at low concentrations. However, as the concentration increases and surpasses the antioxidant threshold, this defense system becomes overwhelmed and ultimately fails [73]. Among the four types of microplastics tested, PS exhibited the highest toxicity. This may be attributed to the rapid formation of a protein corona, a layer of adsorbed proteins on the PS surface upon contact with biological systems, which enhances its biological reactivity and toxicity [74]. Moreover, the relatively high surface charge and polarity of PS particles may increase their affinity for cell membranes and biomolecules, facilitating cellular uptake and prolonging retention time in organisms [75]. These interactions can stimulate ROS production, disrupt membrane integrity, and promote lipid peroxidation [76].

Degradable microplastics can undergo hydrolysis or biodegradation in soil to form soluble organic substances such as oligomers or monomers [77], thereby increasing soil organic carbon. And the increase in biodegradable microplastics is significantly greater than that of traditional non-biodegradable microplastics [78]. However, in this study, as shown in Figure 5B, the soil organic matter content in the PP and PS stress groups was significantly higher than that in the PLA and PHA stress groups. However, the cascading coupling effect value was significantly higher in the PLA and PHA stress groups than in the PP and PS stress groups. On the one hand, this may be due to the “substrate competition” between the degradation process of biodegradable microplastics and the organic matter mineralization process mediated by earthworms, which amplifies the “obstacles” to both parties. Due to the release of a large amount of carbon sources that are easily utilized by microorganisms during the degradation process of biodegradable plastics, their changes in microbial community turnover and structure are stronger than those of traditional plastics [79,80], leading to the gradual succession of microbial communities to predominantly eutrophic microorganisms (r strategy) [81]. As a typical countermeasure organism, earthworms compete with soil microbial communities for high-energy and easily utilizable substrates, especially when resources are limited or external stress occurs.

On the other hand, it may be caused by the toxic effects of additives in plastic products. Although the ultimate goal of biodegradable microplastics is to mineralize plastic products into carbon dioxide and water, commercially produced PLA and PHA plastic products typically contain additives (such as plasticizers, nucleating agents, masterbatch, and antioxidants) that gradually release into the surrounding environment as the plastic degrades. Biodegradable microplastics can affect the physiological and biochemical processes of organisms, causing growth inhibition, oxidative stress, and inducing cell damage [82]. In this study (Appendix C), the path analysis model showed that the remaining path coefficients were all less than 0.25, and the model validation passed. This indicates that there is a close cascade coupling effect between oxidative stress indicators in earthworms and soil total carbon content after the occurrence of microplastic stress. Further research found (Figure 5C) that the highest cumulative pathway effect coefficient of the PLA stress group was 1.328, while the PHA stress group could reach 1.648. Both were much higher than the PP and PS stress groups (the former was 0.904; the latter was 0.971). This also confirms that biodegradable microplastics are not friendly to soil ecosystems. In the future, it is necessary to further evaluate their ecological toxicity and focus on studying the aging, degradation, release of intermediate products and additives of biodegradable microplastics in soil.

## 4. Conclusions

This study is the first to integrate the oxidative stress effect of microplastics on *Eisenia fetida* with the competition between the degradation of biodegradable microplastics and earthworm-mediated organic mineralization for substrates, and construct a joint pathway analysis model of biological physiological responses and soil ecological functions. This model realizes the cross-scale correlation between individual physiological toxicity and ecosystem function, providing a new framework for quantifying the comprehensive harm of microplastics and filling the gap in the physiological–ecological linkage in the ecological safety assessment of degradable plastics.

The stage pattern of oxidative stress markers, SOD, POD, and CAT, are warning markers in the early stage (day 7); POD, GST, and GPX dominate free radical scavenging and cell repair in the middle stage; and SOD activates the immune–detoxification–neural synergistic mechanism to cope with long-term stress in later stages, providing a basis for temporal dynamic assessment. The toxicity mechanism of different microplastic types, PLA, PHA, PP, and PS, all cause significant damage to the antioxidant defense system and cell membrane integrity of earthworms, but PLA relies on POD to clear peroxides and PP relies on GST for detoxification. PLA- and PS-induced nerve damage occurs in the early stages of exposure, while the neurotoxic effects of PHA manifest in later stages of stress. This thus provides key evidence revealing the long-term risks of biodegradable plastics. Risk confirmation of biodegradable microplastics: although PLA and PHA are biodegradable, they inhibit earthworm-mediated organic mineralization through substrate competition and cause oxidative stress and nerve damage, confirming that they are not environmentally friendly, especially in soil ecosystems.

In summary, these findings provide new insights into the different toxicity mechanisms of traditional and biodegradable microplastics, and provide a scientific basis to evaluate the ecological risks of microplastic pollution in soil and whether biodegradable plastics are truly environmentally friendly.

## Figures and Tables

**Figure 1 antioxidants-14-01197-f001:**
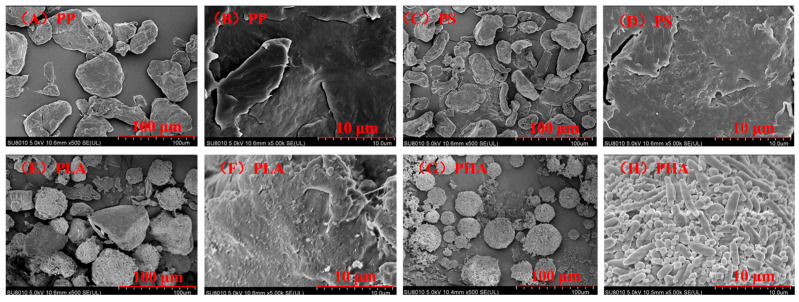
SEM images of four types of microplastics ((**A**,**C**,**E**,**G**): 500; (**B**,**D**,**F**,**H**): 5000).

**Figure 2 antioxidants-14-01197-f002:**
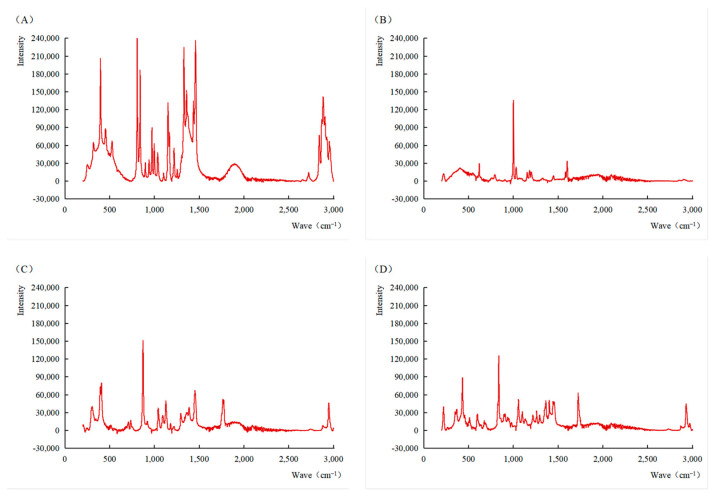
Raman spectra of four types of microplastics. ((**A**): PP; (**B**): PS; (**C**): PLA; (**D**): PHA).

**Figure 3 antioxidants-14-01197-f003:**
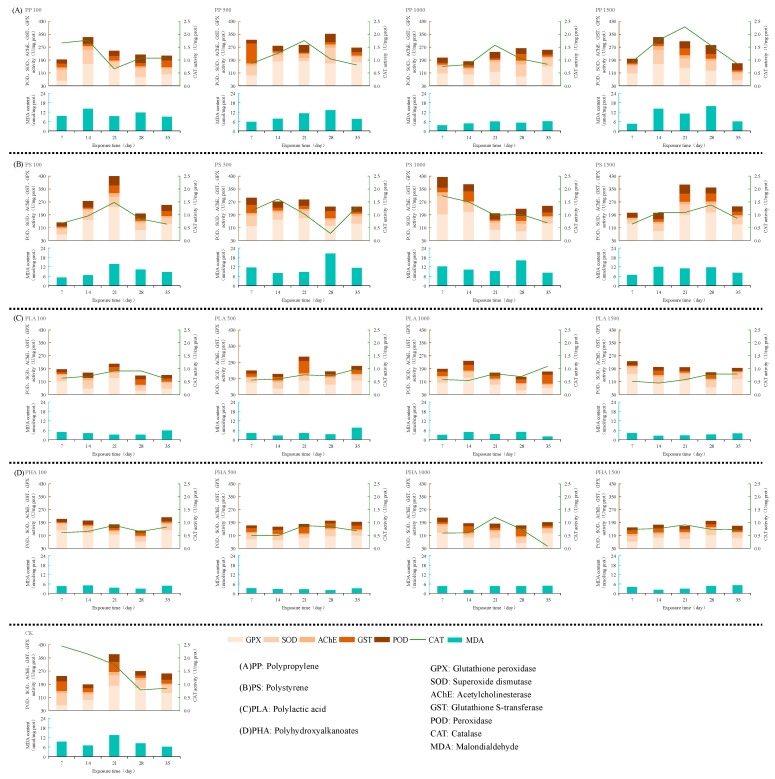
Oxidative stress effect of earthworm under four kinds of microplastic stress.

**Figure 4 antioxidants-14-01197-f004:**
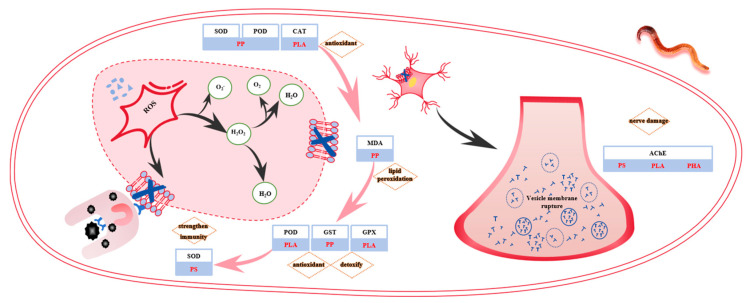
Mechanism diagram of the antioxidant system and nervous system of earthworms under microplastic stress.

**Figure 5 antioxidants-14-01197-f005:**
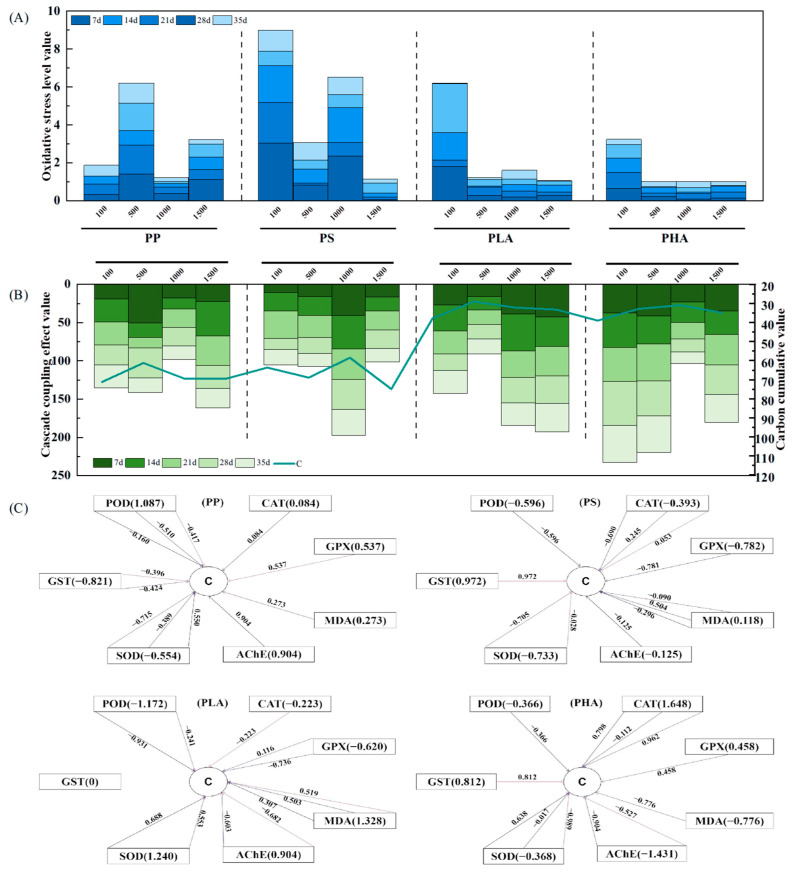
Toxicity of four types of microplastics on earthworms and soil ((**A**): the degree of oxidative stress in earthworms; (**B**): cascade coupling effect diagram; (**C**): correlation coefficient graph of earthworm oxidative stress indicators and soil total).

**Table 1 antioxidants-14-01197-t001:** Screening of sensitive oxidative stress indicators at different stress stages.

Stress Test Group	Stress Time Day				
	7	14	21	28	35
PP	SOD	MDA	POD	MDA	GST
PS	POD	AChE	SOD	SOD	AChE
PLA	CAT	GPX	POD	POD	POD
PHA	POD	POD	AChE, MDA	AChE	SOD

## Data Availability

The sequencing data have been uploaded to the Science Data Bank. If readers need to the data, they can click on the link: https://www.scidb.cn/s/ZBVfmm (accessed on 7 March 2025).

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
