# Peer review of "Reassessing Whether Biodegradable Microplastics Are Environmentally Friendly: Differences in Earthworm Physiological Responses and Soil Carbon Function Impacts"

_antioxidants, 2025, doi:10.3390/antiox14101197_

Round 1

Reviewer 1 Report

Journal: Antioxidants

Review comments for Article entitled:

Reassessing the Ecological Friendliness of Biodegradable Microplastics: Differences from Earthworm Physiological Responses and Soil Carbon Function Impacts

This is an interesting and useful study aimed to evaluate the oxidative stress responses of the earthworm species Eisenia fetida, when exposed to the various types of microplastics: polypropylene (PP), polystyrene (PS), polylactic acid (PLA), and polyhydroxyalkanoates (PHA). The authors applied a progressive factor analysis principal component analysis model in order to pinpoint sensitive oxidative stress biomarkers in earthworms subjected to microplastic exposure. The findings of this research are significant, offering valuable experimental data that could serve as a reference for ecological risk assessments related to microplastic pollution in soil ecosystems. Furthermore, this study not only contributes to the scientific community's understanding of microplastic impacts on terrestrial ecosystems, but also emphasizes the need for sustainable practices in managing plastic waste.

General comments:

The Introduction is well written, in a clear, precise, and systematic manner. The research background is clearly presented. The gap of knowledge, as well as the aim of the study are clearly presented and explained.

The experimental designs, sample size, as well as statistical analyses are appropriate, and the data collection, procession, and interpretation are reproducible based on the details given in the Materials and Methods section.

The obtained results are also clearly presented, discussed in systematic manner, presented in one table and five figures in the main text and compared with previously published results. The authors provided four appendices (A, B, C and D) clearly explained.

The Conclusions are well written, and represent the core of your research and the obtained results.

Specific comments:

In order to improve the quality of this manuscript I would suggest the following:

  • Line 22: Please delete “and” at the beginning of the sentence.
  • Line 25: Please delete “and” at the beginning of the sentence. I suggest that you start the sentence with “Furthermore, this study provide…”.
  • Lines 82-84: This sentence belongs to the section Conclusions.
  • Lines 129-130: Please rephrase the sentence in order to describe the process, like in previous sentences. I suggest using passive voice.
  • Lines: 142-143: Please rephrase the sentence as it is not clearly connected to the previous text.
  • Line 196: Is this sentence a subtitle? “Select the principal component.” I suggest rephrasing it as: “Selection of the principal components.”
  • Line 198: Please change “are” to “were”.
  • Line 202: Please consider inserting “:” after the word “formulas”.
  • Line 210: Please change both “is” to “was”.
  • Lines 209-217: The paragraph is not clearly written. Please rephrase it. I suggest the following: “After the occurrence of microplastic stress, the oxidative stress index in earthworms was used as the independent variable Xi and the total carbon content of soil was the dependent variable Y. A path analysis model was constructed in order to analyze the relationship between biological physiological responses and soil ecological functions in microplastic polluted environments, where: rij…..”
  • Line 216: I suggest this form at the beginning of the sentence: “Additionally, rij was decomposed…”
  • Line 289: Please change “induce” into “induced”.
  • Line 362: Please change the title of the Figure 4 in more detailed way. For example, “Mechanism” – of what?
  • Chapter 3.3. I could not find the citation of reference with number 60, although it is stated in the list of references.
  • Lines 467-468: Please review the supplementary materials in form of numbers / letters. You stated here Appendix 1 and Appendix 2, and in supplementary material Appendix A, B, C, D… Please add the titles of appendices.

English language (grammar and spelling) is fine, minor revision is needed.

Author Response

  1. Comment:The Introduction is well written, in a clear, precise, and systematic manner. The research background is clearly presented. The gap of knowledge, as well as the aim of the study are clearly presented and explained.The experimental designs, sample size, as well as statistical analyses are appropriate, and the data collection, procession, and interpretation are reproducible based on the details given in the Materials and Methods section.The obtained results are also clearly presented, discussed in systematic manner, presented in one table and five figures in the main text and compared with previously published results. The authors provided four appendices (A, B, C and D) clearly explained.The Conclusions are well written, and represent the core of your research and the obtained results.

Response: Thank you very much for reviewing this manuscript. At the same time, thank you very much for your constructive comments and suggestions which would help us in depth to improve the quality of this manuscript.

  1. Comment:Please delete “and” at the beginning of the sentence

Response: Thank you very much for your comments.We agree with this opinion. Therefore, we have deleted “and”.(Line 22)

  1. Comment:Please delete “and” at the beginning of the sentence. I suggest that you start the sentence with “Furthermore, this study provide…”.

Response: Thank you very much for your comments.We agree with this opinion. Therefore, we have changed the sentence to “Furthermore, this study provides direct evidence for proving that biodegradable microplastics lack ecological friendliness by breaking through the one-way research framework of "microplastic biotoxicity" and innovatively constructing a path analysis model that links biological physiological responses with soil ecological functions. ”.(Line 22-26)

  1. Comment:This sentence belongs to the section Conclusions

Response: Thank you very much for your comments.We agree with this opinion. Therefore, we have removed this sentence and added it to the conclusion (Line 498-501)

  1. Comment:Please rephrase the sentence in order to describe the process, like in previous sentences. I suggest using passive voice.   

Response: Thank you very much for your comments.We agree with this opinion. Therefore, we have changed the sentence to “10 g soil was taken from each bottle every 7 days, air-dried it naturally, ground it and then sieved it through a 200 mesh sieve for the determination of total soil carbon.” (Line 139-141)

  1. Comment:Please rephrase the sentence as it is not clearly connected to the previous text.   

Response: Thank you very much for your comments.We agree with this opinion. Therefore, we have changed the sentence to “ Soil total carbon (TC) was measured using combustion method.” (Line 153-154)

  1. Comment:Is this sentence a subtitle? “Select the principal component.” I suggest rephrasing it as: “Selection of the principal components.”        

Response: Thank you very much for your comments.This sentence is not a subtitle, it only described the modeling process, but we have also made modifications to it by changing it to “Selection of the principal component. ” (Line 205)

  1. Comment:Please change “are” to “were”.      

Response: Thank you very much for your comments.We agree with this opinion. Therefore, We have changed “are” to “were” (Line 207)

  1. Comment: Please consider inserting “:” after the word “formulas”.     

Response: Thank you very much for your comments.We agree with this opinion. Therefore, We have added “:” (Line 210)

  1. Comment: Please change both “is” to “was”.  

Response: Thank you very much for your comments.We agree with this opinion. Therefore, We have changed “is” to “was” (Line 218)

  1. Comment:The paragraph is not clearly written. Please rephrase it. I suggest the following: “After the occurrence of microplastic stress, the oxidative stress index in earthworms was used as the independent variable Xi and the total carbon content of soil was the dependent variable Y. A path analysis model was constructed in order to analyze the relationship between biological physiological responses and soil ecological functions in microplastic polluted environments, where: rij…..”  

Response: Thank you very much for your comments.We agree with this opinion. Therefore, We have changed this paragraph to “After the occurrence of microplastic stress, the oxidative stress index in earthworms is used as the independent variable and the total carbon content of soil was the dependent variable . A path analysis model was constructed in order to analyze the relationship between biological physiological responses and soil ecological functions in microplastic polluted environments, where:” (Line 218-221)

  1. Comment: I suggest this form at the beginning of the sentence: “Additionally, rij was decomposed…”

Response: Thank you very much for your comments.We agree with this opinion. Therefore, We have added 'additively' at the beginning of the sentence and changed “can be” to “was” (Line 225)

  1. Comment:Please change “induce” into “induced”.

Response: Thank you very much for your comments.We agree with this opinion. Therefore, We have changed “induce” to “induced” (Line 306)

  1. Comment:Please change the title of the Figure 4 in more detailed way. For example, “Mechanism” – of what?

Response: Thank you very much for your comments.We agree with this opinion. Therefore, We have provided a detailed description of the mechanism:Mechanism Diagram of the Antioxidant System and Nervous System of Earthworms Under Microplastic Stress (Line 382-383)

  1. Comment:Chapter 3.3. I could not find the citation of reference with number 60, although it is stated in the list of references.

Response: Thank you very much for your comments.We agree with this opinion. Therefore, We have added reference number 62 in the correct location. (Line 416)

  1. Comment:Please review the supplementary materials in form of numbers / letters. You stated here Appendix 1 and Appendix 2, and in supplementary material Appendix A, B, C, D… Please add the titles of appendices.

Response: Thank you very much for your comments.We agree with this opinion. Therefore, The title in the supplement has been changed to “1, 2, 3, 4” and the missing explanation has been added. (Line 527, Line 540, Line 570, Line 579,Line 503-504)

Reviewer 2 Report

Your manuscript entitled “Reassessing the Ecological Friendliness of Biodegradable Microplastics: Differences from Earthworm Physiological Responses and Soil Carbon Function Impacts” is interesting. However, something needs to be addressed before it is submitted again.

Line 21: How did you measure neurotoxicity?

Line 32: You must improve the introduction.

Line 38. This effect is dependent on the concentration of microplastic.

Line 72: What mathematical models?

Line 237: You can leave only 5000

Line 239: You must improve figure 2 quality as well as other figures.

Line 272: You must improve figure size.

Line 427: What is the name of the Y axis?

Conclusion is like a summary; you have to improve.

The manuscript is interesting. There are a lot of work.

Author Response

  1. Comment:Reassessing the Ecological Friendliness of Biodegradable Microplastics: Differences from Earthworm Physiological Responses and Soil Carbon Function Impacts” is interesting. However, something needs to be addressed before it is submitted again.

Response: Thank you very much for reviewing this manuscript, and thanks for your excellent and professional revision. We have studied the comments carefully and made corrections which we hope meet with approval.

  1. Comment:How did you measure neurotoxicity?

Response: Thanks for reviewing this manuscript. We used acetylcholinesterase activity(AChE) to evaluate neurotoxicity, as described in lines 353-353 of the article (Line 22).

  1. Comment:You must improve the introduction.(Line32)

Response: Thanks for reviewing this manuscript. We have provided detailed information on the excellent properties of plastic “their low production cost, high plasticity, and stable chemical properties” (Line 34-35).

  1. Comment:This effect is dependent on the concentration of microplastic.

Response: Thanks for reviewing this manuscript. We have provided detailed explanations. “With the long-term and continuous accumulation of these microplastics in the soil, they will not only have a negative impact  on soil structure, hinder nutrient cycling, reduce water retention capacity, and pose serious threats to soil biota , making them a pressing global environmental concern.” (Line 40-42).

  1. Comment:What mathematical models?

Response: Thanks for reviewing this manuscript. We have added specific mathematical models: “a combined model of principal component analysis and factor analysis” (Line 76-77).

  1. Comment:You can leave only 5000

Response: Thanks for reviewing this manuscript.We have removed “×” and only kept “500, 5000”(Line 244).

  1. Comment:You must improve figure 2 quality as well as other figures.

Response: Thanks for reviewing this manuscript.We have improved the clarity. (Line 243 Line 245 Line 289 Line 381 Line 451).

  1. Comment:You must improve figure size.

Response: Thanks for reviewing this manuscript.We have improved figure size. (Line 289).

  1. Comment:What is the name of the Y axis?

Response: Thanks for reviewing this manuscript. We have changed the Y-axis to A: Oxidative stress level value; B: Cascade coupling effect value, Carbon cumulative value (Line 451).

  1. Comment:Conclusion is like a summary; you have to improve.

Response: Thanks for reviewing this manuscript. We have revised the conclusion. (Line 474-501).

Reviewer 3 Report

The authors tested several oxidative stress markers in the compost earthworm Eisenia fetida exposed for up to 35 days to high concentrations of microplastics: two conventional (PP and PS) and two biodegradable (PLA and PHA) polymer types. The topic is important and timely, and the authors seem to have put efforts in the preparation of this manuscript.

I can however not recommend it for publication, unless several corrections and clarifications are made. The main area of uncertainty is how the various responses under MP exposures are different from the natural variation observed in the control treatment (Figure 3). This should be more clearly discussed. In addition, the abstract and conclusions are not necessarily supported by the results: It is indeed not possible to infer consequences on the "haplic phaeozem ecosystem" from oxidative stress markers in one earthworm species exposed in the laboratory. 

The list below is by no means exhaustive, especially when language is considered.

l.86: Microplastics cannot be referred to as "reagents"

l.87: "200 mesh powders": what does this mean? What is the particle size distribution of the particles? MP are clearly not characterized enough.

l.94: where is that?

l.95: 1 (or 2) digit for pH, not 3.

l.99: w:w or v:v ratio?

l.99: Weird statement in parentheses, this need to be supported by analysis.

l.112: Ref.18: what type of ref is this?

l.116: already stated

l.117: not “pre treated”, depurated instead

l.129-130: Language

l.132: You probably mean +4°C.

l.142-143: Language

l.145-219: Data analysis is beyond my expertise, so someone else needs to review this part.

l.223-224: SEM and Raman spectroscopy should have been introduced earlier in Materials and Methods. The scale (µm) needs to be legible on the micrographs. Particle size distribution of the various MP needs to be stated (at least mean ± SD, number of particles measured).

Fig.2: Without comparison with standards it is difficult to ascertain that these spectra fit those of reference PP, PS, PLA, PHA materials.

Fig.3: The figure is unfortunately not legible (too small). There is no explanation of the graph CK in the legend, but I assume it refers to control earthworms. How are the oxidative stress responses under the various MP exposures statistically different from those in the control treatment?

Fig.5: Not legible 

l.254-257: Not clear at what PP concentration(s) this is happening. How is this different from the control treatment?

l.262-263: This makes no sense: First GST does not peak at day 14 at 1500 mg/kg, secondly the day by which it is supposed to drop is not stated. And again, how is this different from controls?

l.264-266: Not supported by Figure 3.

l.267-271: Not supported by Figure 3.

l.278-279: where are the data supporting this statement?

From this point on, I lost trust in the way data were reported/discussed and will not comment in detail here. 

Author Response

  1. Comment:The authors tested several oxidative stress markers in the compost earthworm Eisenia fetida exposed for up to 35 days to high concentrations of microplastics: two conventional (PP and PS) and two biodegradable (PLA and PHA) polymer types. The topic is important and timely, and the authors seem to have put efforts in the preparation of this manuscript. I can however not recommend it for publication, unless several corrections and clarifications are made. The main area of uncertainty is how the various responses under MP exposures are different from the natural variation observed in the control treatment (Figure 3). This should be more clearly discussed. In addition, the abstract and conclusions are not necessarily supported by the results: It is indeed not possible to infer consequences on the "haplic phaeozem ecosystem" from oxidative stress markers in one earthworm species exposed in the laboratory. 

Response: Thanks for reviewing this manuscript.We added the differences in various reactions of earthworms exposed to microplastics compared to the control group and summarized the conclusions again. (Line 264-266 Line 285-288 Line 474-501) The Eisenia fetida we have selected is a model organism recommended by ISO, OECD, and GB/T for ecotoxicological testing, with the main purpose of using this organism to indicate the toxic effects of pollutants. The analysis of multiple earthworm species is also a method commonly used among researchers. Similarly, our team has conducted research among different species (Ge Y, Pesticide Biochemistry and Physiology, 2025, 214, 106607; Ge Y, Pesticide Biochemistry and Physiology, 2025, 207, 106215). However, in this study, given the wide variety of pollutants involved, when we incorporated mathematical modeling, we found that differences among species could not only reduce the scientific validity of the analytical results but also lead to redundancy in the analysis process. Therefore, in this manuscript, we have opted for a more appropriate way to present our findings.

  1. Comment:Microplastics cannot be referred to as "reagents"

Response: Thanks for reviewing this manuscript. We have separated the characterization of microplastics, so we have not changed “reagents”(Line 98)

  1. Comment:"200 mesh powders": what does this mean? What is the particle size distribution of the particles? MP are clearly not characterized enough.

Response: Thanks for reviewing this manuscript.We have increased the particle size of microplastics to 75μm ± 10μm with a mesh size of 200.(Line 90-91)

  1. Comment:where is that?

Response: Thanks for reviewing this manuscript. But we didn't find 'that' in that line or even in that paragraph.This paragraph mainly discussed the sources and preparation methods of the soil used in the experiment

  1. Comment:1 (or 2) digit for pH, not 3.

Response: Thanks for reviewing this manuscript.We have changed to keep two decimal places. (Line 105)

  1. Comment:w:w or v:v ratio?

Response: Thanks for reviewing this manuscript.We have added “w/w”. (Line 109)

  1. Comment:Weird statement in parentheses, this need to be supported by analysis.

Response: Thanks for reviewing this manuscript. We have added the original content of microplastics, lead, cadmium, and nickel. (Line 109-110)

  1. Comment:18: what type of ref is this?

Response: Thanks for reviewing this manuscript.We have indicated that this thesis belongs to the master's thesis category. (Line 619)

  1. Comment:18:already stated

Response: Thanks for reviewing this manuscript.We have deleted this sentence.

  1. Comment:not “pre treated”, depurated instead

Response: Thanks for reviewing this manuscript.We have changed “pre treated” to “depurated” (Line 127)

  1. Comment:Language

Response: Thanks for reviewing this manuscript.We have revised this sentence as “10 g soil was taken from each bottle every 7 days, air-dried it naturally, ground it and then sieved it through a 200 mesh sieve for the determination of total soil carbon.” (Line 139-140)

  1. Comment:You probably mean +4°

Response: Thanks for reviewing this manuscript.We have changed to “+4 °C” (Line 143)

  1. Comment:Language

Response: Thanks for reviewing this manuscript.We have revised this sentence as Soil total carbon (TC) was measured using combustion method .”(Line 153-154)

  1. Comment:Data analysis is beyond my expertise, so someone else needs to review this part.

Response: Thanks for reviewing this manuscript.Perhaps the mathematical analysis model used in this study exceeded the professional scope of the reviewing experts.

  1. Comment:SEM and Raman spectroscopy should have been introduced earlier in Materials and Methods. The scale (µm) needs to be legible on the micrographs. Particle size distribution of the various MP needs to be stated (at least mean ± SD, number of particles measured).

Response: Thanks for reviewing this manuscript.We have added an introduction to scanning electron microscopy and Raman spectroscopy related methods in the Materials and Methods section; The microscopic image clearly indicates the scale (unit: μm, micrometers); The particle size of microplastics has also been marked.(Line 89-97 Line 244)

  1. Comment:2: Without comparison with standards it is difficult to ascertain that these spectra fit those of reference PP, PS, PLA, PHA materials.

Response: Thanks for reviewing this manuscript.We apologize that the qualitative analysis of common microplastics such as polypropylene (PP), polystyrene (PS), polylactic acid (PLA), and polyhydroxyalkanoates (PHA) is based on the specific peak combinations determined by their molecular structures. Standard spectrum comparison is not a necessary step. (Line 231-242)

  1. Comment:3: The figure is unfortunately not legible (too small). There is no explanation of the graph CK in the legend, but I assume it refers to control earthworms. How are the oxidative stress responses under the various MP exposures statistically different from those in the control treatment?

Response: Thanks for reviewing this manuscript.We have adjusted the image clarity and added the oxidative stress response of different microplastic exposure groups compared to the control group. (Line 264-266 Line 285-288)

  1. Comment:5: Not legible 

Response: Thanks for reviewing this manuscript.We have adjusted the clarity. (Line 451)

  1. Comment:Not clear at what PP concentration(s) this is happening. How is this different from the control treatment?

Response:Thanks for reviewing this manuscript.We have specified the specific concentration of polypropylene (PP) at the time of the above phenomenon and the differences compared to the control group. (Line 260-266)

  1. Comment:This makes no sense: First GST does not peak at day 14 at 1500 mg/kg, secondly the day by which it is supposed to drop is not stated. And again, how is this different from controls?

Response:Thanks for reviewing this manuscript.We have rewritten this paragraph and added a comparison with the control group. (Line 270-272 Line 285-288)

  1. Comment:Not supported by Figure 3(Line 264-266)

Response:Thanks for reviewing this manuscript. Perhaps the mathematical analysis model used in this study exceeds the background knowledge of some review experts in the professional field, which may lead to a certain deviation in understanding the research results. Nevertheless, we have optimized and revised the content of this paragraph. (Line 278-284)

  1. Comment:Not supported by Figure 3(Line 267-271)

Response:Thanks for reviewing this manuscript.Same as above 21.(Line 278-284)

  1. Comment:where are the data supporting this statement?

Response:Thanks for reviewing this manuscript.We have added support for reference [43]. (Line 297)